# Seek Commonality but Preserve Differences: Dissected Dynamics Modeling for Multi-modal Visual RL

**Yangru Huang[1], Peixi Peng[2,3]\*, Yifan Zhao[4], Guangyao Chen[1], Yonghong Tian[1,2,3]\***

[1]School of Computer Science, Peking University
[2]School of Electronic and Computer Engineering, Shenzhen Graduate School, Peking University
[3]Peng Cheng Laboratory
[4]School of Computer Science and Engineering, Beihang University,
`yrhuang@stu.pku.edu.cn`,
`{pxpeng, gy.chen, yhtian}@pku.edu.cn`, `zhaoyf@buaa.edu.cn`

## Abstract

Accurate environment dynamics modeling is crucial for obtaining effective state representations in visual reinforcement learning (RL) applications. However, when facing multiple input modalities, existing dynamics modeling methods (*e.g.*, Deep-MDP) usually stumble in addressing the complex and volatile relationship between different modalities. In this paper, we study the problem of efficient dynamics modeling for multi-modal visual RL. We find that under the existence of modality heterogeneity, modality-correlated and distinct features are equally important but play different roles in reflecting the evolution of environmental dynamics. Motivated by this fact, we propose Dissected Dynamics Modeling (DDM), a novel multi-modal dynamics modeling method for visual RL. Unlike existing methods, DDM explicitly distinguishes consistent and inconsistent information across modalities and treats them separately with a divide-and-conquer strategy. This is done by dispatching the features carrying different information into distinct dynamics modeling pathways, which naturally form a series of implicit regularizations along the learning trajectories. In addition, a reward predictive function is further introduced to filter task-irrelevant information in both modality-consistent and inconsistent features, ensuring information integrity while avoiding potential distractions. Extensive experiments show that DDM consistently achieves competitive performance in challenging multi-modal visual environments. The code is available in this link: https://github.com/Yara-HYR/DDM

## 1 Introduction

Recent years have witnessed a growing interest in visual reinforcement learning (RL). By leveraging raw pixel inputs directly from sensory data, visual RL allows agents to navigate and interpret their environments with unprecedented precision and adaptability [33, 24, 51]. Since raw pixels are usually redundant and suffer from expanded data dimensionality, a key aspect of visual RL is to acquire efficient and compact state representations. A typical approach involves modeling environmental dynamics in an abstracted state space [1], where the state representations are supposed to mimic the actual environmental changes but operate on a sufficiently low dimension. One classical solution is DeepMDP [9], which leverages neural networks to compress high-dimensional input data. Given consecutive observations, DeepMDP predicts reward and state transitions within the deep latent space, enabling more effective policy optimization over complex, dynamic environments.

---

\*Corresponding author

38th Conference on Neural Information Processing Systems (NeurIPS 2024).

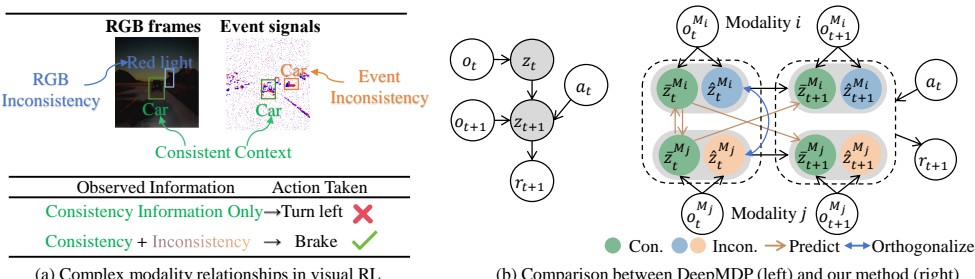

(a) Complex modality relationships in visual RL

(b) Comparison between DeepMDP (left) and our method (right)

Figure 1: Motivation of our method and comparison with DeepMDP [9] designed for single modality. $o_t$, $z_t$, $a_t$, and $r_t$ represent the observation, latent feature, action, and reward at time $t$, respectively. $\overline{z}_t^{M_i}$ and $\hat{z}_t^{M_i}$ are the decoupled consistent (Con.) and inconsistent (Incon.) features for $i$-th modality.

Although methods like DeepMDP have proven effective for environmental dynamics modeling, research on this subject has been mostly restricted to the single visual modality case (*e.g.*, with only RGB frame pixels as input). However, under multi-modal scenarios, different modalities may share correlated scene descriptions but meanwhile preserve their own unique observations. In addition, such a modality relationship is not stable but constantly evolves with the highly dynamic environment. An intuitive example is given in Fig. 1 (a), where we show the observations of both RGB frames and event camera signals [26, 3] on a night-time driving scene. It can be seen that the two modalities only exhibit partially consistent context. Meanwhile, their inconsistencies include critical visual clues, such as traffic lights (in RGB frames) and cars hidden in the dark (in event signals). Existing approaches often overlook this complex and volatile status caused by modality heterogeneity, which may have detrimental impacts on learning correct environmental dynamics. A straightforward solution might be to model the dynamics of each modality separately. However, this approach neglects the complementarity between different data sources, which could reduce its effectiveness.

To achieve efficient dynamics modeling in multi-modal scenarios, decoupling and understanding the interplay between modalities and environmental dynamics is necessary. *Firstly*, modality-correlated features provide a foundational perspective by capturing shared and common information across different sensory inputs. These commonalities are critical for building a cohesive description of the environment's overall behaviour. *Secondly*, the distinct features in each modality offer inconsistent side views of the environment. In conventional multi-modal visual tasks, these inconsistencies are typically deemed less critical and are mitigated through modality alignment [5, 40, 27]. However, in modeling RL dynamics, the inconsistent features are equally important as they 1) carry critical task-related clues and 2) are also the key ingredient in deducing the modality-specific state changes. This importance is also verified by our investigation presented in Sec. 4.4, where we find that incorporating modality inconsistencies leads to a more accurate mimicry of the environment rewards.

Based on the above analysis, we propose **Dissected Dynamics Modeling (DDM)**, a method that neatly integrates the decomposition of commonalities and differences across modalities with the dynamics modeling process. As shown in Fig. 1(b), DDM seeks a mutual predictive property between modalities to establish an implicit regularization and isolate modality-consistent features. Furthermore, it extends this consistency across different temporal locations, ensuring that the learned common information resonates with environmental dynamics. Regarding modality inconsistencies, unlike standard practice that enforces mutual exclusivity with consistent features of the same modality, DDM imposes orthogonal constraints between different modalities to highlight their unique contents. This avoids overly strong regularization, which leads to improved feature quality. The decomposed inconsistent features are only used to establish the dynamics of each modality itself, which creates a more complete environment dynamics without causing interference across modalities. Despite dissecting modality content and model dynamics separately, DDM further introduces a reward predictive function to ensure that both consistent and inconsistent features focus on task-relevant information. This approach preserves information integrity while avoiding potential distractions, which ensures optimized decision-making accuracy and responsiveness.

In summary, our work contributes threefold: 1) We analyze the challenges of modeling multi-modal environment dynamics for visual RL, offering insights into the key factors that influence modeling robustness. 2) We present DDM, a novel method that seeks commonality but preserves differences

between modalities for enhanced dynamics modeling. 3) Our experimental results validate the effectiveness of our method, demonstrating its strength under complex environmental conditions.

## 2 Related Work

**Visual Reinforcement Learning.** Visual reinforcement learning (RL) aims to enable agents to make decisions based on raw visual inputs. Existing methods typically revolve around strategies such as data augmentation to enhance input diversity [50, 29, 30, 31], incorporating auxiliary tasks for richer representation supervision [23, 53, 19, 35, 46], employing world models for behavior prediction [14, 13, 38, 49], or pre-training encoders to improve state representation compression [28, 39, 51]. Despite these advances, current visual RL methods predominantly rely on the RGB modality, which could potentially limit the agent's holistic understanding of the environment.

**Multi-modal Reinforcement Learning.** Handling multi-modal input is important for various control tasks [18, 44, 22] but remains as an under-explored research area in the field of RL. Despite the advancements in multi-modal machine learning methods, applying these techniques to RL faces several challenges due to the highly dynamic RL environment and heterogeneous modalities. A few pioneering works have focused explicitly on multi-modal RL [4, 32, 20]. However, these methods primarily focus on aligning different modalities, using techniques like mutual information optimization [4] or imposing consistency constraints [32]. Such a strategy overlooks the impact of modality-inconsistent aspects on decision-making, which may hinder its effectiveness.

**Dynamics Modeling in Visual RL.** Learning to model the environment dynamics in visual RL is critical for obtaining efficient state representations and reducing computational complexity. Earlier works involve reconstructing input observations at the pixel level [48, 45]. Subsequent studies have focused on dynamics modeling by predicting future latent states [10, 9, 14, 37, 25, 2, 41], showing promising results. Although these methods have advanced the field, they are not explicitly designed for multi-modal scenarios and may face challenges under complex modality relationships. Unlike existing methods, our approach uniquely dissects modality contents based on their relationships, treating consistent and inconsistent information separately within the dynamics modeling process.

## 3 Methodology

We consider the standard RL setting where an agent interacts with the environment in consecutive time steps. At each time step, the agent has access to observations from $d$ distinct modalities, where $d > 1$. Fig. 2 illustrates the proposed Dissected Dynamics Modeling (DDM) method. DDM is structured around two tightly entangled elements: 1) the decomposition of commonalities and differences across modalities and 2) modeling modality-aware environmental dynamics. Through this modeling process, the decomposed features of each modality are optimized to retain highly abstract and comprehensive task-related information. These features are then applied to learn policies, which promote both decision-making precision and sample efficiency. We now describe our overall approach in detail.

### 3.1 Preliminaries

**Multi-modal Markov Decision Process.** The task of multi-modal visual RL can be formulated as a Markov Decision Process with the tuple $(\mathcal{S}, \mathcal{A}, \mathcal{T}, \mathcal{R}, \gamma)$, where the state space is $\mathcal{S} = \prod_{i=1}^{d} \mathcal{O}^{M_i}$ represented by the combined observation space comprising $d$ different modalities [20]. $\mathcal{A}$ is the action space, $\mathcal{T}(s_{t+1}|s_t, a_t)$ is the transition function, $\mathcal{R}(s, a)$ is the reward function, and $\gamma \in [0, 1)$ is the discount factor [42]. At every time step $t$, the agent observes all modality data to take action according to policy $\pi$. The environment then returns a reward to the agent. The goal is to optimize the policy by maximizing the expected cumulative reward.

**Soft Actor Critic.** Our approach is based on Soft Actor-Critic (SAC) [11, 12], which iteratively refines a policy function $\pi$ and a critic function $Q$. SAC seeks to maximize the expected cumulative reward while simultaneously encouraging exploration by an $\alpha$-discounted maximum entropy:

$$\mathcal{L}_\pi = \mathbb{E}_{a_t \sim \pi} \left[ Q(o_t, a_t) - \alpha \log \pi(a_t|o_t) \right], \tag{1}$$

The parameters of the value function are updated by the Bellman equation:

$$\mathcal{L}_Q = \mathbb{E}_{(o_t, a_t) \sim \mathcal{D}} \left[ (Q(o_t, a_t) - (r_t + \gamma V(o_{t+1})))^2 \right], \tag{2}$$

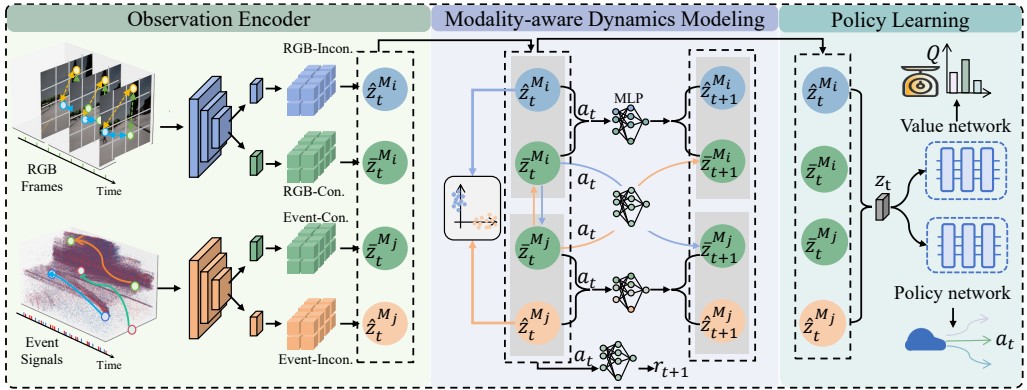

Figure 2: The proposed Dissected Dynamics Modeling (DDM) method. The input visual modalities are first passed through separate observation encoders to get partitioned modality-consistent and inconsistent features. These features then undergo modality-aware dynamics modeling to model accurate RL dynamics and promote feature quality. Finally, the optimized modality features are merged as state representations, which are then used for robust policy learning.

where $D$ is the replay buffer, and the soft state value function $V(o_{t+1})$ is calculated as follows:

$$V(o_{t+1}) = \mathbb{E}_{\tilde{a}_{t+1} \sim \pi} \left[ \bar{Q}(o_{t+1}, \tilde{a}_{t+1}) - \alpha \log \pi(\tilde{a}_{t+1}|o_{t+1}) \right], \tag{3}$$

where $\bar{Q}$ is the exponential moving average of the critic network $Q$, and $\tilde{a}_{t+1}$ is the action produced by the current policy.

## 3.2 Modality Decomposition

**Feature Encoding and Partitioning.** Given the significant disparities in low-level information inputs across different modalities, we utilize $d$ modality-specific encoders to transform high-dimensional observations into deep latent features. These encoders possess identical network architectures, except for the initial input layer, which is adjusted to match the channel number of the corresponding input data shapes. The extracted feature of the $i$-th modality $M_i$ at time step $t$ is denoted as $z_t^{M_i}$. Since our method aims to decompose each modality into common and unique components, we further adopt two separate network branches as information filters for feature partitioning. These two branches process the extracted modality feature $z_t^{M_i}$, producing two distinct outputs: $\overline{z}_t^{M_i}$ as the modality-consistent feature, and $\hat{z}_t^{M_i}$ as the modality-inconsistent feature. In the subsequent decomposition phase, these features undergo a series of regularizations to ensure they accurately convey the intended information. For clarity, we mainly introduce the case involving two modalities. The extension to multiple modalities will be discussed subsequently.

**Extracting Consistencies Between Modalities.** Consistent information across modalities provides foundational knowledge about general environmental behavior. To extract such information, we focus on enhancing the mutual predictive properties between different modalities. Given two modalities $M_1$ and $M_2$, we first impose a mutual prediction (mp) constraint on the partitioned modality-consistent features $\overline{z}_t^{M_1}$ and $\overline{z}_t^{M_2}$ at the current time step $t$:

$$\mathcal{L}_{mp} = \|F_1(\overline{z}_t^{M_2}) - \overline{z}_t^{M_1}\| + \|F_2(\overline{z}_t^{M_1}) - \overline{z}_t^{M_2}\|, \tag{4}$$

where $F_1$ and $F_2$ are prediction heads implemented as Multi-Layer Perceptrons (MLP). Such a prediction between modalities triggers a form of content alignment, offering immediate common environmental insights. However, merely imposing the consistency constraint at a static step $t$ may not fully leverage the pivotal temporal dynamics of the environment. Intuitively, observations in adjacent timesteps often preserve similar contexts. For example, in autonomous driving, a car that exists in one timestep will likely remain in the next. Meanwhile, there may be subtle variations in details, such as the car's location, distance, and size. These temporal differences can serve as augmented views that facilitate the discrimination of modality commonalities. Therefore, we create an additional transition prediction constraint across different modalities. Given $\overline{z}_t^{M_1}$, $\overline{z}_t^{M_2}$ and the

action $a_t$ taken in time step $t$, we set up two transition prediction (tp) heads $G_1$, $G_2$ to predict the consistent features $\overline{z}_{t+1}^{M_1}$, $\overline{z}_{t+1}^{M_2}$ of the next time step:

$$\mathcal{L}_{tp} = \|G_1(\overline{z}_t^{M_2}, a_t) - \overline{z}_{t+1}^{M_1}\|^2 + \|G_2(\overline{z}_t^{M_1}, a_t) - \overline{z}_{t+1}^{M_2}\|^2. \tag{5}$$

This approach creates a synergistic effect on both consistency extraction and dynamics modeling: on one hand, the expanded view in the subsequent time step offers enhanced guidance for learning to identify task-relevant common objects. On the other hand, predicting environmental transitions improves feature quality related to modeling scene dynamics. This interconnection tightly couples the two tasks, allowing them to reinforce each other effectively.

**Identifying Inconsistencies.** Extracting inconsistent information between different modalities is a subtler task than consistency extraction. This subtleness is caused by the absence of clear constraints on feature content: unlike consistency extraction, which involves pulling features closer with concrete objective functions, inconsistency extraction lacks such a definitive goal. For initial consideration, any content that is not consistent across modalities should be deemed as inconsistencies. Therefore, a common practice is to enforce mutual exclusivity between the consistent and inconsistent features of the same modality [21]. Taking modalities $M_1$ and $M_2$ as examples, the orthogonality objective can be applied to impose this mutual exclusivity as follows:

$$\mathcal{L} = (\overline{z}_t^{M_1} \cdot \hat{z}_t^{M_1})^2 + (\overline{z}_t^{M_2} \cdot \hat{z}_t^{M_2})^2, \tag{6}$$

where $(\cdot)$ denotes the dot product operation. However, this objective might be overly strong, as Eq. 6 forces the feature pairs $[\overline{z}_t^{M_1}, \hat{z}_t^{M_1}]$ and $[\overline{z}_t^{M_2}, \hat{z}_t^{M_2}]$ to be completely uncorrelated. Since each pair of features originated from the same modality encoder, they are likely to share similar encoding patterns. Therefore, enforcing orthogonality may lead to excessively departed features and decrease feature expressiveness. To alleviate this issue, we take an alternative solution. Given two modalities $M_1$ and $M_2$, we instead optimize the following objective:

$$\mathcal{L}_{orth} = (\hat{z}_t^{M_1} \cdot \hat{z}_t^{M_2})^2. \tag{7}$$

Eq. 7 directly matches our intention of highlighting different contexts across modalities. During optimization, the similar contexts in $M_1$ and $M_2$ are gradually excluded in $\hat{z}_t^{M_1}$ and $\hat{z}_t^{M_2}$ due to the strict constraint of orthogonality. Meanwhile, this optimization process is less aggressive than Eq. 6 since $\hat{z}_t^{M_1}$ and $\hat{z}_t^{M_2}$ come from different modality encoders and minimizing their correlation does not affect learning effective representations within each encoder.

**Extension to Multiple Modalities.** So far we have been working on the two-modality case. The extension to more modalities is also straightforward. For consistency extraction, we modify the prediction heads $F_i$ and $G_i$ for modality $M_i$ to take all other modalities as input. The objectives in Eq. 4 and 5 then become:

$$\mathcal{L}_{mp} = \sum_{i=1}^{d} \|F_i(\{\overline{z}_t^{M_j} \mid j \neq i\}) - \overline{z}_t^{M_i}\|, \mathcal{L}_{tp} = \sum_{i=1}^{d} \|G_i(\{\overline{z}_t^{M_j} \mid j \neq i\}, a_t) - \overline{z}_{t+1}^{M_i}\|^2, \tag{8}$$

where $d > 2$ is the modality number and $\{\overline{z}_t^{M_j} \mid j \neq i\}$ denotes all $d-1$ modality-consistent features except for $\overline{z}_t^{M_i}$. These $d-1$ features are concatenated and sent into $F_i$ and $G_i$ for prediction. Eq. 8 is inspired by set calculation, where each feature is regarded as a set of contexts. It states that for $d$ sets, if the union of any $d-1$ sets equals the remaining set, then all $d$ sets must be identical and consistent. For inconsistency extraction, we simply extend Eq. 7 as:

$$\mathcal{L}_{orth} = \sum_{i=1}^{d-1} \sum_{j=i+1}^{d} (\hat{z}_t^{M_i} \cdot \hat{z}_t^{M_j})^2, \tag{9}$$

which covers all pair-wise modality combinations.

## 3.3 Modality-aware Dynamics Modeling

After decomposing modalities into consistent and inconsistent contexts, our approach treats them separately in the overall dynamics modeling process. The consistency features are used to predict cross-modality future states to ensure they capture common scene dynamics, as formulated by the

Table 1: Comparison with state-of-the-art methods on CARLA benchmark. S-RL denotes single-modality RL methods, M-CV denotes multi-modal methods for conventional computer vision tasks, and M-RL denotes multi-modal RL methods. ER represents the episode return, and D(m) is the driving distance in meters. The best results are **bolded**.

| Methods | Type | Normal | | Midnight | | Dazzling | | Rainy | | Average | |
|---|---|---|---|---|---|---|---|---|---|---|---|
| | | ER | D(m) | ER | D(m) | ER | D(m) | ER | D(m) | ER | D(m) |
| SAC | S-RL | 204 ± 49 | 253 ± 35 | 154 ± 58 | 207 ± 35 | 125 ± 57 | 181 ± 39 | 174 ± 42 | 217 ± 34 | 164.25 | 214.50 |
| DrQ | S-RL | 234 ± 43 | 272 ± 32 | 195 ± 41 | 248 ± 25 | 132 ± 53 | 179 ± 42 | 225 ± 56 | 258 ± 41 | 196.50 | 239.25 |
| DeepMDP | S-RL | 241 ± 39 | 286 ± 40 | 210 ± 23 | 244 ± 29 | 162 ± 35 | 223 ± 32 | 252 ± 42 | 267 ± 37 | 216.25 | 255.00 |
| SPR | S-RL | 256 ± 45 | 297 ± 53 | 205 ± 67 | 232 ± 48 | 165 ± 62 | 220 ± 49 | 261 ± 53 | 280 ± 52 | 221.75 | 257.25 |
| TransFuser | M-CV | 249 ± 73 | 286 ± 60 | 224 ± 58 | 267 ± 40 | 178 ± 58 | 209 ± 43 | 248 ± 70 | 274 ± 57 | 224.75 | 259.00 |
| EFNet | M-CV | 254 ± 67 | 309 ± 54 | 239 ± 60 | 264 ± 43 | 174 ± 57 | 212 ± 38 | 259 ± 66 | 276 ± 48 | 231.50 | 265.25 |
| MuMMI | M-RL | 233 ± 70 | 297 ± 58 | 215 ± 59 | 258 ± 39 | 159 ± 65 | 206 ± 51 | 228 ± 61 | 261 ± 49 | 208.75 | 255.50 |
| MAIE | M-RL | 241 ± 58 | 291 ± 43 | 217 ± 61 | 242 ± 41 | 163 ± 58 | 210 ± 45 | 242 ± 52 | 274 ± 40 | 215.75 | 254.25 |
| HAVE | M-RL | 275 ± 77 | 315 ± 63 | 243 ± 75 | 263 ± 45 | 189 ± 68 | 237 ± 52 | 275 ± 67 | 286 ± 50 | 245.50 | 275.25 |
| Ours-DDM | M-RL | **289 ± 61** | **338 ± 52** | **279 ± 41** | **300 ± 46** | **229 ± 42** | **267 ± 47** | **294 ± 35** | **314 ± 42** | **272.75** | **304.75** |

$\mathcal{L}_{tp}$ in Eq. 5 and 8. The inconsistent contents are sent into a different pathway, which is used to deduce the full modality-specific evolution. In particular, for each modality $M_i$, we first merge its consistent and inconsistent features into a unified modality representation:

$$\tilde{z}_t^{M_i} = \bar{z}_t^{M_i} + \hat{z}_t^{M_i}, \tag{10}$$

the merged $\tilde{z}_t^{M_i}$ contains a comprehensive description of $M_i$. Then, we set up a full-state predictive (fp) head $P_i$ for each $M_i$, which takes both $\tilde{z}_t^{M_i}$ and action $a_t$ to forecast the next state:

$$\mathcal{L}_{fp} = \sum_{i=1}^{d} \|P_i(\tilde{z}_t^{M_i}, a_t) - \tilde{z}_{t+1}^{M_i}\|^2. \tag{11}$$

Such a full-state prediction ensures the combination of decomposed modality features faithfully captures the modality-specific dynamics. To accurately reflect environmental rewards, a reward predictive function $R$ is further introduced to filter task-irrelevant information, optimized by:

$$\mathcal{L}_r = \|R(z_t, a_t) - r_{t+1}\|, \tag{12}$$

where $z_t = \left[ \tilde{z}_t^{M_1}, \tilde{z}_t^{M_2}, \ldots, \tilde{z}_t^{M_d} \right]$ is the ultimate state representation obtained by concatenating features from all $d$ modalities, and $r_{t+1}$ is the actual reward returned by the environment.

### 3.4 Policy-learning with DDM

Given the feature decomposition and dynamics modeling processes, the policy-learning of our dissected dynamics modeling (DDM) method involves the joint optimization of multiple objectives that are divided into three groups:

$$\mathcal{L}_{\text{DDM}} = \underbrace{\mathcal{L}_Q + \mathcal{L}_\pi}_{\text{SAC}} + \underbrace{\mathcal{L}_{mp} + \mathcal{L}_{orth}}_{\text{Feat. Decomposition}} + \underbrace{\mathcal{L}_{tp} + \mathcal{L}_{fp} + \mathcal{L}_r}_{\text{Dynamics Modeling}}, \tag{13}$$

where $\mathcal{L}_{tp}$ is used in both feature decomposition and dynamics modeling, and the SAC takes the state representation $z_t$ as input to replace the observation $o_t$. During training, the DDM learns to extract effective state representations that faithfully reflect the environmental behaviour, yielding improved policy in various multi-modal visual RL tasks.

## 4 Experiments

### 4.1 Experiment Settings

**Environments.** To evaluate the effectiveness of our approach in complex multi-modal scenarios, we conducted tests on two widely used RL benchmarks: 1) the CARLA simulator [7] for autonomous driving, and 2) the DeepMind Control (DMControl) suite [43] for robotic control. For CARLA, we assessed our method under four different environmental conditions: "Normal", "Midnight", "Dazzling", and "Rainy". The latter three conditions provide challenging driving scenarios characterized by

Table 2: Comparing with state-of-the-art methods on DMControl. The best results are **bolded**.

| Methods | Type | Cartpole Swingup | Reacher Easy | Cheetah Run | Average | Cartpole Swingup_sparse | Reacher Hard | Quadruped Run | Average |
|---------|------|------------------|--------------|-------------|---------|-------------------------|--------------|---------------|---------|
| SAC | S-RL | 742 ±60 | 307 ±79 | 204 ±94 | 417.67 | 711 ±42 | 45 ±32 | 204 ±37 | 320.00 |
| DrQ | S-RL | 815 ±58 | **913 ±54** | 239 ±75 | 655.67 | 674 ±67 | 105 ±45 | 469 ±46 | 416.00 |
| DeepMDP | S-RL | 827 ±66 | 823 ±89 | 524 ±40 | 724.67 | 713 ±51 | 128 ±32 | 453 ±42 | 431.33 |
| SPR | S-RL | 845 ±49 | 905 ±37 | 581 ±45 | 777.00 | 720 ±54 | 187 ±48 | 468 ±39 | 458.33 |
| TransFuser | M-CV | 845 ±56 | 906 ±48 | 518 ±52 | 756.33 | 728 ±42 | 169 ±51 | 211 ±75 | 369.33 |
| EFNet | M-CV | 824 ±43 | 900 ±35 | 555 ±34 | 759.67 | 769 ±55 | 125 ±57 | 183 ±81 | 359.00 |
| MuMMI | M-RL | 802 ±59 | 895 ±57 | 530 ±54 | 742.33 | 712 ±60 | 152 ±43 | 328 ±77 | 397.33 |
| MAIE | M-RL | 829 ±60 | 852 ±52 | 535 ±61 | 738.67 | 750 ±37 | 147 ±58 | 402 ±69 | 433.00 |
| HAVE | M-RL | 835 ±52 | 867 ±65 | 560 ±59 | 754.00 | 732 ±45 | 151 ±69 | 426 ±58 | 436.33 |
| Ours-DDM | M-RL | **854 ±46** | 904 ±42 | **587 ±32** | **781.67** | **813 ±42** | **224 ±64** | **489 ±56** | **508.67** |

extreme lighting and heavy rainfall, closely mimicking real-world driving scenes. Various modalities provided by the benchmark are utilized, including RGB frames, event signals, depth images, and LiDAR images in a bird's-eye view (BEV) perspective. For DMControl, we conduct evaluations on six tasks, which are partitioned into two groups based on their difficulty levels. The modalities used are RGB frames and depth images. However, given the relatively clean backgrounds of tasks in DMControl and the high consistency between the two modalities, we additionally introduce inconsistencies by randomly selecting one modality and masking out 20% of its image contents. This intentional introduction of inconsistencies simulates the occlusion issues commonly encountered during real-world machine operations, thereby increasing the task difficulty.

**Compared Methods.** We compare our DDM method with various types of methods, including single-modal baseline RL algorithms SAC [12] and DrQ [50], two competitive dynamics modeling methods DeepMDP [9] and SPR [37], conventional multi-modal methods EFNet [40] and TransFuser [5], and multi-modal RL methods MUMMI [4], MAIE [32], and HAVE [20]. For EFNet and TransFuser, since they were initially designed for traditional computer vision tasks, we mainly evaluate their core modality fusion and alignment modules. These modules are integrated with the SAC algorithm for decision-making, which is the same base RL algorithm used in our method.

**Implementation Details.** Our approach is implemented with Pytorch. To ensure a fair comparison, we use the same encoder network architectures and training hyperparameters for all methods being evaluated. In line with established practices [50, 23], we convert data from each modality into image-based representations and stack multiple sequential images to capture temporal dynamics. Methods on the CARLA benchmark and DMControl are trained for 100k and 500k frames, respectively. To obtain reliable results, we train each task-method pair for 5 runs and report the mean values and standard deviations after 20 evaluations. More detailed experimental setups are given in the Appendix.

## 4.2 Comparison with State-of-the-art

**Results on CARLA.** Table 1 presents the performance of various methods on the CARLA benchmark using two modalities: RGB frames and event signals. The results demonstrate that our method outperforms others in all four environmental conditions. The table shows that the performances of the SAC and DrQ baselines are less satisfactory, especially in complex conditions such as "Dazzling" and "Midnight". While conventional dynamics modeling methods like DeepMDP and SPR do enhance performance, their improvements are modest compared to our DDM method. For example, DDM obtains an average episode return improvement of approximately 24.4% and 21.3% over DeepMDP and SPR, respectively. This substantial enhancement is attributed to the explicit modality decomposition and processing in DDM, which significantly boosts robustness across diverse multi-modal environments. The table also indicates that existing multi-modal methods achieve decent performance on CARLA. However, these methods primarily focus on modality alignment and fusion, neglecting inconsistency contexts critical for decision-making. Consequently, they do not match the efficiency of our DDM, which effectively leverages both consistent and inconsistent modality contexts for robust decision-making. The results with all four modalities are given in the Appendix.

**Results on DMControl.** We further evaluate different methods on the DMControl in Table 2. Similar to the results on the CARLA benchmark, our method demonstrates significant performance improvements on DMControl. It is apparent that conventional multi-modal methods such as TransFuser and EFNet exhibit limited performance, likely due to the increased inconsistency brought by random masking. In contrast, our DDM method consistently enhances the performance of the SAC baseline by a large margin, further verifying its effectiveness.

Table 3: Ablation study of different components in DDM on CARLA benchmark.

| Methods | Normal | | Midnight | | Dazzling | | Rainy | | Average | |
|---|---|---|---|---|---|---|---|---|---|---|
| | ER | D(m) | ER | D(m) | ER | D(m) | ER | D(m) | ER | D(m) |
| RGB | 169 ± 66 | 213 ± 49 | 121 ± 54 | 187 ± 41 | 84 ± 62 | 123 ± 53 | 159 ± 58 | 194 ± 41 | 133.25 | 179.25 |
| Baseline | 204 ± 49 | 253 ± 35 | 154 ± 58 | 207 ± 35 | 125 ± 57 | 181 ± 39 | 174 ± 42 | 217 ± 34 | 164.25 | 214.50 |
| $+\mathcal{L}_{fp}$ | 234 ± 64 | 275 ± 52 | 198 ± 63 | 232 ± 48 | 157 ± 52 | 199 ± 50 | 241 ± 53 | 274 ± 45 | 207.50 | 245.00 |
| $+\mathcal{L}_{r}$ | 249 ± 48 | 287 ± 54 | 212 ± 47 | 248 ± 42 | 164 ± 66 | 224 ± 48 | 254 ± 47 | 271 ± 39 | 219.75 | 257.50 |
| $+\mathcal{L}_{mp}$ | 262 ± 51 | 312 ± 39 | 227 ± 54 | 260 ± 59 | 184 ± 59 | 243 ± 47 | 258 ± 54 | 282 ± 45 | 232.75 | 274.25 |
| $+\mathcal{L}_{tp}$ | 277 ± 59 | 324 ± 44 | 248 ± 58 | 272 ± 52 | 208 ± 46 | 254 ± 43 | 279 ± 47 | 298 ± 40 | 253.00 | 287.00 |
| $+\mathcal{L}_{orth}$ | **289 ± 61** | **338 ± 52** | **279 ± 41** | **300 ± 46** | **229 ± 42** | **267 ± 47** | **294 ± 35** | **314 ± 42** | **272.75** | **304.75** |

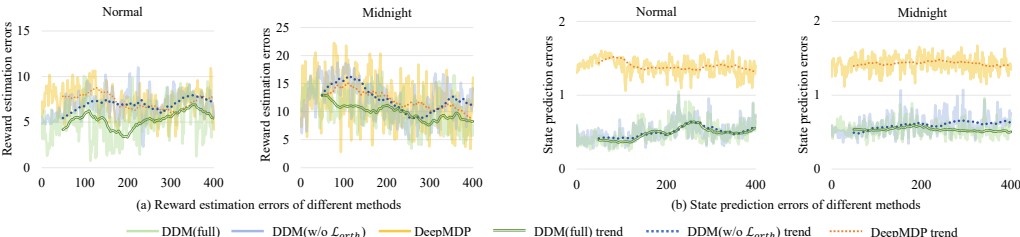

Figure 3: The reward estimation and state prediction errors of different methods.

## 4.3 Ablation Study

To systematically verify the effectiveness of each component in DDM, we conduct a series of ablation experiments on the CARLA benchmark. The experiments are conducted by progressively integrating different training objectives in DDM into the baseline SAC algorithm. The results are given in Table 3. Compared with the baseline, incorporating the full-state prediction objective $\mathcal{L}_{fp}$ and reward prediction objective $\mathcal{L}_r$ notably enhances performance. However, this enhancement does not bring a significant advantage over conventional methods such as DeepMDP and SPR reported in Table 1, since the modality relationship is not explicitly modeled. Further applying $\mathcal{L}_{mp}$ and $\mathcal{L}_{tp}$ to model modality consistencies provides additional performance gains, showing the importance of extracting common contexts in multi-modal RL. Finally, the integration of $\mathcal{L}_{orth}$ to model modality inconsistencies yields substantial improvement, confirming the effectiveness of our modality decomposition strategy.

## 4.4 Discussion and Analysis

**The Importance of Modality Commonality and Differences.** A key finding of our study is the crucial role of both modality commonality and differences in multi-modal dynamics modeling. To investigate this, we leverage the bisimulation metrics [8, 9] for MDPs to assess the dynamics modeling performance of different methods. Specifically, the bisimulation metrics define two states to be behaviourally similar if they have close rewards along the transition trajectory [9]. Therefore, the difference between the estimated and true rewards can reflect how closely the learned states mimic the actual environmental states. In Fig. 3(a), we plot the absolute reward difference $|R(z_t, a_t) - r_{t+1}|$ for three methods: DeepMDP, DDM without using $\mathcal{L}_{orth}$ (*i.e.*, only modeling modality consistency), and the full DDM model. The reward difference is plotted for each time step of the same test sequence on the CARLA benchmark. From the figure, it is clear that compared to DeepMDP, modeling modality consistency can reduce reward estimation errors. Additionally, incorporating modality inconsistency further significantly enhances estimation accuracy. These results confirm the importance of both common and differing modality contexts in accurately learning dynamics.

Besides rewards, we also examine the state prediction accuracy of different methods. In Fig. 3(b), we plot the normalized latent state prediction error $\frac{||z'_t - z_{t+1}||}{||z_{t+1}||}$ for each method. For DDM methods, $z'_t$ is defined as $[P_1(\tilde{z}_t^{M_1}, a_t), P_2(\tilde{z}_t^{M_2}, a_t), \ldots, P_d(\tilde{z}_t^{M_d}, a_t)]$, representing the predicted states from multiple modalities, for DeepMDP, $z'_t$ represents the predicted single latent state. The figure shows that DeepMDP exhibits significantly higher state prediction errors, highlighting the benefits of decomposing modality contexts and treating each modality separately. Among the two DDM variants, the full DDM model with $\mathcal{L}_{orth}$ has less prediction error, again demonstrating the advantages of incorporating task-related inconsistencies.

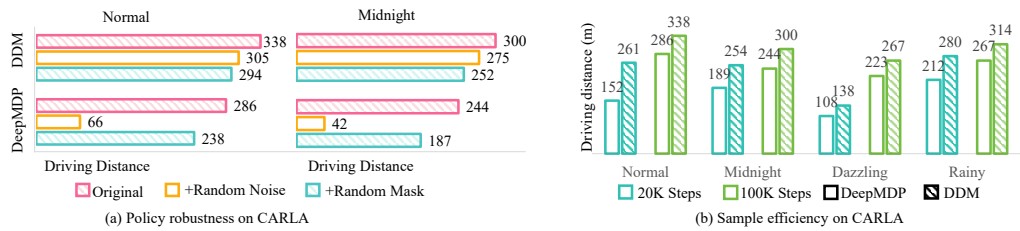

(a) Policy robustness on CARLA

(b) Sample efficiency on CARLA

Figure 4: Comparison between DeepMDP and DDM on policy robustness and sample efficiency.

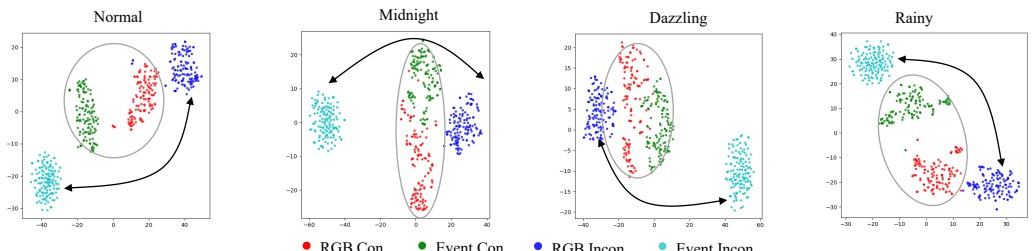

Figure 5: t-SNE visualization of consistent and inconsistent features.

**Policy Robustness and Sample Efficiency.** The modality decomposition and reward prediction in DDM efficiently extract task-related contexts and filter out irrelevant content, potentially enhancing policy robustness against distractions like data noise. To verify this, we evaluate DDM and Deep-MDP on the CARLA benchmark with random Gaussian noises or random masks added to the input modality data. The results in Fig. 4(a) show that DDM experiences minimal performance degradation, highlighting its robustness. In contrast, DeepMDP exhibits a significant performance decline, particularly when faced with random input noises. Despite policy robustness, we further investigate the sample efficiency of the two methods. The results in Fig. 4(b) show that DDM converges faster and achieves better accuracy than DeepMDP in various environmental conditions. These advantages highlight the superiority of DDM, demonstrating its potential for learning diverse visual control tasks.

**Comparison of Different Approaches for Inconsistency Extraction.** In Sec. 3.2, we have discussed two different approaches for inconsistency extraction, *i.e.*, applying the mutual exclusive constraint within the same modality (Eq. 6) or across different modalities (Eq. 7). We conjecture that Eq. 6 might be overly strong, potentially affecting feature expressiveness. To verify this conjecture, we compare the performance of the two inconsistency extraction approaches on the CARLA benchmark. The results in Fig. 7 show that forcing orthogonality within the same modality using Eq. 6

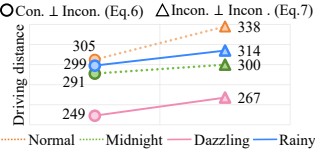

Figure 7: Comparing different inconsistency constraints.

consistently obtains worse performance, indicating that it learns less expressive features compared to our approach.

**Visualization of Decomposed Modality Features.** To confirm that the modality features decomposed by DDM accurately capture the intended contexts, we use t-SNE to visualize both modality-consistent and inconsistent features for RGB and event signals on the CARLA benchmark. As shown in Fig.5, DDM successfully aligns the modality-consistent features, resulting in closer feature distances, while the modality-inconsistent features are distinctly separated. Further visualizations in Fig.6 reveal that the consistent features primarily focus on common objects (e.g., road fences), whereas the inconsistent features highlight objects visible in one modality but not in others (e.g., lane lines in RGB frames). In contrast, DeepMDP tends to indiscriminately focus on all scene objects, lacking contextual discriminativity. These visualizations demonstrate the efficacy of the modality decomposition in DDM, facilitating more precise dynamics modeling and efficient state representation.

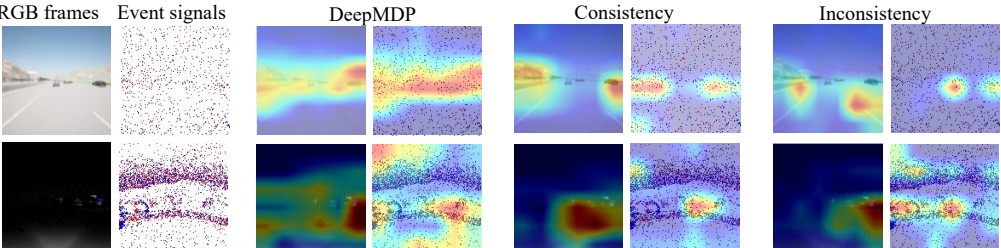

Figure 6: Visualization of feature heatmaps in DDM under normal and midnight conditions.

## 5 Limitations and Future Works

Our current DDM method has two primary limitations. First, while it effectively simulates multi-modal dynamics and enhances representation learning, it lacks integrated planning mechanisms similar to those in MuZero [36] and Dreamer [13, 15, 16, 34, 6]. Research has shown that planning mechanisms can effectively model the impact of various strategies in data-rich scenarios, thus minimizing trial and error in real-world applications. In future works, we plan to integrate long-term action consequences into our decision-making processes by developing decomposed multi-modal world models. Second, our approach focuses primarily on vision-based data inputs. However, recent studies suggest that integrating diverse modal inputs, such as textual and audio data, can yield adaptable representations [47]. Therefore, extending our approach to include additional types of modal inputs represents another promising future direction to enhance its scope and performance.

## 6 Conclusion

We have studied the impact of modality heterogeneity on dynamics modeling in multi-modal visual reinforcement learning (RL) tasks. Our investigation reveals that both the commonalities and differences in each modality are crucial for accurate dynamics modeling, albeit playing distinct roles. Based on this insight, we have presented Dissected Dynamics Modeling (DDM), a novel method designed to enhance dynamics modeling and representation learning for multi-modal visual RL. Our approach represents one of the first explorations of modeling environmental dynamics in multi-modal scenarios, providing a fresh context decomposition perspective based on modality relationships. Extensive experiments demonstrate the effectiveness of our approach, which significantly improves representation quality and decision-making performance in diverse environmental conditions.

## 7 Acknowledgment

The study was funded by the National Natural Science Foundation of China under contracts No. 62332002, No. 62425101, No. 62422602, No. 62372010, No. 62202010, No. 62402015, and the China Postdoctoral Science Foundation under grant 2024M750100.

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

# Appendix

The contents of this Appendix are presented as follows:

Section A offers extra experiment findings, including performance on the CARLA benchmark with more modalities, experiments for visual inputs from multiple camera positions, sample efficiency and different masking ratios on the DeepMind Control Suite.

Section B describes the experimental setups, including the details of network structures, environment conditions, training hyper-parameters, and hardware details respectively.

Section C discusses the potential societal impacts of our method.

Table A1: Comparison with state-of-the-art methods on CARLA benchmark with four modalities.

| Methods | Type | Normal | | Midnight | | Dazzling | | Rainy | | Average | |
|---|---|---|---|---|---|---|---|---|---|---|---|
| | | ER | D(m) | ER | D(m) | ER | D(m) | ER | D(m) | ER | D(m) |
| SAC | S-RL | 182 ± 74 | 282 ± 71 | 198 ± 82 | 246 ± 74 | 174 ± 79 | 211 ± 78 | 194 ± 75 | 250 ± 68 | 187.00 | 247.25 |
| DrQ | S-RL | 229 ± 78 | 329 ± 68 | 218 ± 76 | 262 ± 76 | 206 ± 82 | 231 ± 80 | 239 ± 76 | 287 ± 72 | 223.00 | 277.25 |
| DeepMDP | S-RL | 244 ± 64 | 327 ± 52 | 235 ± 67 | 285 ± 62 | 215 ± 70 | 242 ± 68 | 248 ± 61 | 272 ± 58 | 235.50 | 281.50 |
| SPR | S-RL | 261 ± 71 | 362 ± 73 | 227 ± 83 | 299 ± 74 | 200 ± 86 | 229 ± 75 | 259 ± 84 | 302 ± 70 | 236.75 | 298.00 |
| TransFuser | M-CV | 244 ± 80 | 340 ± 65 | 249 ± 74 | 315 ± 69 | 234 ± 76 | 255 ± 68 | 264 ± 72 | 308 ± 64 | 247.75 | 304.50 |
| EFNet | M-CV | 268 ± 74 | 356 ± 60 | 240 ± 69 | 312 ± 67 | 225 ± 78 | 249 ± 64 | 259 ± 74 | 301 ± 72 | 248.00 | 304.50 |
| MuMMI | M-RL | 249 ± 74 | 334 ± 65 | 223 ± 78 | 300 ± 66 | 227 ± 72 | 239 ± 64 | 254 ± 80 | 299 ± 61 | 238.25 | 293.00 |
| MAIE | M-RL | 254 ± 76 | 341 ± 66 | 234 ± 88 | 311 ± 72 | 238 ± 73 | 248 ± 78 | 260 ± 69 | 300 ± 73 | 246.50 | 300.00 |
| HAVE | M-RL | 295 ± 72 | 365 ± 70 | 275 ± 81 | 334 ± 76 | 258 ± 81 | 267 ± 76 | 307 ± 72 | 337 ± 75 | 283.75 | 325.75 |
| Ours-DDM | M-RL | **316 ± 67** | **378 ± 69** | **296 ± 75** | **355 ± 67** | **276 ± 72** | **282 ± 72** | **324 ± 76** | **353 ± 69** | **303.00** | **342.00** |

Table A2: Results on CARLA benchmark with different perspectives RGB+Lidar-BEV.

| Methods | Normal | | Midnight | | Dazzling | | Rainy | | Average | |
|---|---|---|---|---|---|---|---|---|---|---|
| | ER | D(m) | ER | D(m) | ER | D(m) | ER | D(m) | ER | D(m) |
| HAVE | 241 ± 64 | 282 ± 67 | 251 ± 82 | 285 ± 74 | 221 ± 73 | 244 ± 74 | 272 ± 75 | 281 ± 67 | 246.25 | 273.00 |
| Ours-DDM | 268 ± 72 | 313 ± 55 | 282 ± 63 | 308 ± 46 | 239 ± 64 | 268 ± 59 | 284 ± 55 | 305 ± 60 | 268.25 | 298.50 |

Table A3: Results on DMControl with different perspectives: RGB and depth modality with different camera perspectives .

| Methods | Type | Cartpole Swingup | Reacher Easy | Cheetah Run | Average | Cartpole Swingup_sparse | Reacher Hard | Quadruped Run | Average |
|---|---|---|---|---|---|---|---|---|---|
| SPR | S-RL | 792 ±57 | 839 ±42 | 542 ±49 | 724.33 | 704 ±68 | 134 ±52 | 401 ±58 | 413.00 |
| Ours-DDM | M-RL | 820 ±40 | 887 ±47 | 555 ±61 | 754.00 | 776 ±57 | 182 ±64 | 452 ±65 | 470.00 |

# A  Additional Experimental Results

**Results with More Modalities on CARLA.** Table A1 presents the results of various methods on the CARLA benchmark, utilizing four modalities: RGB frames, event signals, depth images, and LiDAR BEV images. Compared to the results in Table 1, nearly all methods show improved performance with the increased number of modalities. Additionally, the trends observed in Table A1 follow those in Table 1, with single-modal methods typically underperforming their multi-modal counterparts. Among all the multi-modal methods, our DDM method consistently maintains the best performance due to its modality decomposition and integration. This comprehensive modeling allows DDM to effectively utilize additional information provided by multiple modalities, leading to robust decision-making compared to other methods.

**Experiments for visual inputs from multiple camera positions.** We have further verified the ability of our model on multiple camera positions. First, we switch the camera view of RGB modality in DMControl. Second, we test on CARLA with RGB and LiDAR BEV as input modalities. LiDAR BEV is a bird view map, whose perspective is very different from RGB. For simplicity, we only compare our method against the most competitive baseline methods on these two environments (*i.e.,* HAVE on CARLA and SPR on DMControl). The results and illustrations of different camera views are presented in Table A2, Table A3 and Fig. A1 respectively. These results show that our method also works on multiple camera positions.

**Sample efficiency on DMControl.** In Fig. A2, we evaluate the sample efficiency of different methods on DMControl. Compared to DeepMDP and EFNet, our method demonstrates the highest sample

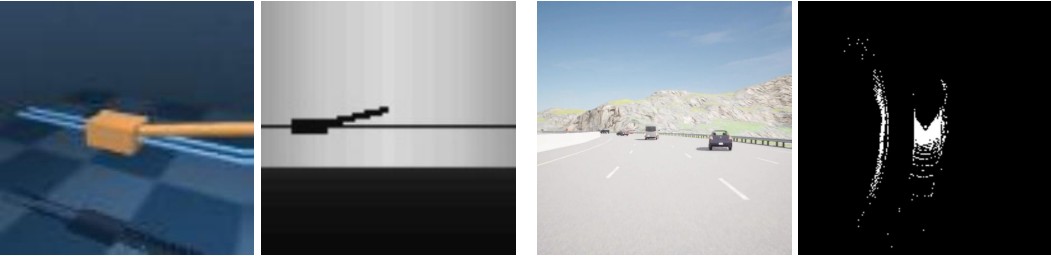

(a) RGB and Depth modalities with different camera perspectives.

(b) RGB and Lidar-BEV modalities with different perspectives on Carla.

Figure A1: Illustration of visual inputs from multiple camera positions.

efficiency across five tasks, as shown in the figure. This superior performance is attributed to our method's ability to capture a more comprehensive representation of the environment. By accurately modeling the dynamics of each modality and their interactions, our method can make more informed decisions with fewer samples, enhancing learning efficiency and overall performance.

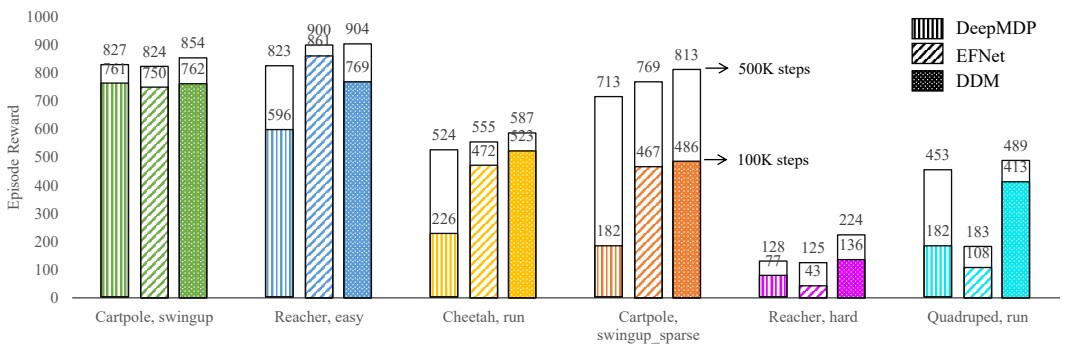

Figure A2: Sample efficiency on DMControl. For each task, we provide the task returns of three methods at 100K and 500K steps, respectively.

Table A4: Different mask ratio on DMControl.

| Methods | Masking Ratio | Cartpole Swingup | Reacher Easy | Cheetah Run | Average | Cartpole Swingup_sparse | Reacher Hard | Quadruped Run | Average |
|---|---|---|---|---|---|---|---|---|---|
| SPR | 0 | 870 ±53 | 956 ±40 | 662 ±49 | 829.33 | 802 ±61 | 564 ±63 | 537 ±58 | 634.33 |
|  | 0.2 | 845 ±49 | 905 ±37 | 581 ±45 | 777.00 | 720 ±54 | 187 ±48 | 468 ±39 | 458.33 |
|  | 0.4 | 832 ±62 | 844 ±59 | 563 ±58 | 746.33 | 687 ±70 | 142 ±68 | 419 ±72 | 416.00 |
|  | 0.6 | 813 ±63 | 824 ±59 | 525 ±65 | 720.67 | 628 ±59 | 82 ±72 | 403 ±68 | 371.00 |
| DDM | 0 | 868 ±42 | 955 ±37 | 676 ±62 | 833.00 | 829 ±52 | 597 ±71 | 552 ±59 | 659.33 |
|  | 0.2 | 854 ±46 | 904 ±42 | 587 ±32 | 781.67 | 813 ±42 | 224 ±64 | 489 ±56 | 508.67 |
|  | 0.4 | 838 ±50 | 852 ±42 | 585 ±70 | 758.33 | 795 ±35 | 211 ±64 | 437 ±62 | 481.00 |
|  | 0.6 | 845 ±75 | 818 ±68 | 524 ±72 | 729.00 | 761 ±80 | 185 ±67 | 446 ±63 | 464.00 |

**Different masking ratios on DMControl .** Our masking operation is performed independently at each timestamp. Therefore, both the masked modality type and the masking locations vary randomly across different timestamps, which simulates a challenging occlusion scenario. For other masking ratios, we further test the ratio of 0%, 40%, 60%, and compare our method with SPR, the most competitive baseline on DMControl. As shown in Table A4, the results reveal that as the masking ratio increases, both SPR and our DDM experience performance drops. However, DDM still outperforms SPR at different ratios.

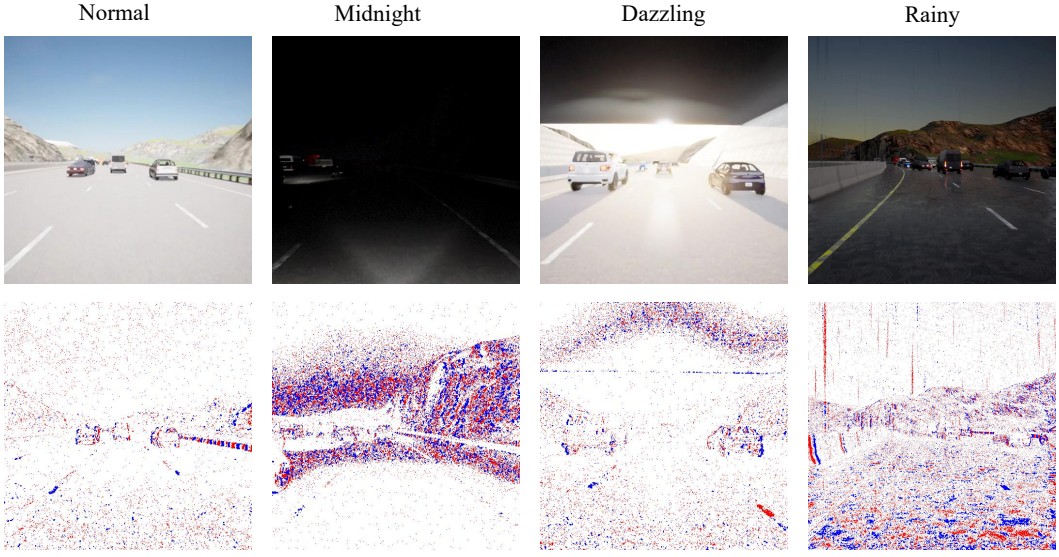

Figure A3: Demonstration of RGB frames and event signals on the CARLA benchmark.

# B Detailed Experimental Setup

In this section, we provide a comprehensive description of the experimental setup, including network architectures, environment configurations, and hyper-parameter settings.

## B.1 Network Architecture

**Modality Encoders.** Individual feature encoders are employed for each modality to handle the input data. Each input observation provided to the encoder consists of 3 consecutive frames returned by the environment. This results in input channel numbers of 9 for RGB frames, 15 for event signals, 3 for LiDAR BEV images, and 3 for depth images. All the feature extractors share a common network architecture that comprises four convolutional layers, each followed by ReLU activations. The final layer produces $64 \times 6 \times 6$ feature maps, which serves as the individual modality features $z_t^{M_i}$. These features are then passed into two separate network branches to get 50-dimensional modality-consistent and inconsistent features $\bar{z}_t^{M_i}$ and $\hat{z}_t^{M_i}$, respectively.

**Actor and Critic Networks.** In our method, the actor network, which is composed of three fully connected (FC) layers with ReLU activations placed after the first two layers, produces parameters for a Gaussian distribution over actions. The critic network has a similar structure but outputs an estimated Q-value for a specific state-action pair.

**Prediction Heads and Reward Prediction Network.** The prediction heads, as described in [9], take the input latent vector and the corresponding action as inputs. They generate the inferred latent vectors through two fully connected (FC) layers, with ReLU activation following the first layer. Similarly, the reward prediction network uses the same architecture but outputs a single-dimensional predicted reward. This network estimates the expected reward for a specific state-action pair, offering critical feedback for dynamics modeling.

## B.2 Environment Details

### B.2.1 Environmental Conditions in CARLA

The four environmental conditions (Normal, Midnight, Dazzling, and Rainy) employed in our study are predefined with specific weather hyper-parameters, as detailed in Table A5. These conditions provide a comprehensive evaluation of the RL algorithms under various situations.

Table A5: The hyper-parameters of weather conditions on CARLA.

| Weather Parameters | Normal | Midnight | Dazzling | Rainy |
|---|---|---|---|---|
| Cloudiness | 5 | 30 | 0 | 0 |
| Precipitation | 0 | 0 | 0 | 100 |
| Precipitation_deposits | 0 | 0 | 0 | 100 |
| wind_intensity | 10 | 0 | 0 | 0 |
| fog_density | 2 | 20 | 0 | 0 |
| fog_distance | 0.75 | 0 | 1000 | 0 |
| wetness | 0 | 0 | 0 | 50 |
| sun_azimuth_angle | -1 | 0 | 270 | 0 |
| sun_altitude_angle | 45 | -90 | 10 | 5 |

### B.2.2 Modality Settings

**Modalities in CARLA.** RGB frames are the default input modality in CARLA, which capture rich texture and color information but may face challenges like motion blur and limited dynamic range under extreme light conditions. Event signal is a new modality provided in CARLA, mimicking the output of neuromorphic event cameras [26, 3]. As shown in Fig. A3, event signals focus on changes in brightness, offering a wider dynamic range, no motion blur, and high temporal resolution, making them ideal for dynamic scenes. However, they do not generate signals at static regions when no pixel differences occur. In addition, event signals may experience hot spot noises. We simulate this noise to replicate real event signal characteristics. Some key parameters related to these two modalities are provided in Table A6. Besides RGB frames and event signals, the depth images in CARLA provide distance maps, while LiDAR BEV creates scanning of the vehicle surroundings. Both depth and LiDAR adopt the default simulation parameters in CARLA.

**Modalities in DMControl.** In DMControl, the RGB frame modality captures direct visual information from the environment, providing detailed texture and color data essential for recognizing objects and their spatial relationships. Complementing this, the depth modality generates a depth map where each pixel value represents the distance to the corresponding point in the scene, offering crucial 3D spatial information. The settings of these two modalities are based on the default parameters.

Table A6: The hyper-parameter settings of the experiments on CARLA.

| RGB | |
|---|---|
| Exposure_speed_up | 3.0 |
| Exposure_speed_down | 1.0 |
| Blur_amount | 1.0 |
| Motion_blur_intensity | 1.0 |
| Motion_blur_max_distortion | 0.8 |
| Motion_blur_min_object_screen_size | 0.4 |
| Lens_flare_intensity | 0.2 |
| Shutter_speed | 100.0 |
| Events | |
| Positive_threshold | 0.2 (Midnight)/0.1(Others) |
| Negative_threshold | 0.2 (Midnight)/0.1(Others) |
| Sigma_positive_threshold | 0.2 |
| Sigma_negative_threshold | 0.2 |
| Other setting of environment | |
| Num_cars | 20 |
| Num_cameras | 1 |

### B.2.3 Reward Setting of CARLA

In the CARLA autonomous driving task, the objective of the agent is to travel as far as it can on a highway without colliding in various weather conditions. Therefore, the reward function is designed similarly to [52] and [20]:

$$r_t = v_{ego}^\top \hat{u}_{highway} \cdot \Delta t - \lambda_c \cdot collision - \lambda_s \cdot |steer| - \lambda_b \cdot brake \qquad (14)$$

The first term is devised to motivate the vehicle to cover as much distance as possible along the highway. $v_{ego}^\top$ indicates the velocity vector of the agent vehicle, $\hat{u}_{highway}$ represents the unit direction vector of the highway, and $\Delta t = 0.1$ stands for the discretized simulation time. The final three terms are incorporated to guarantee that the vehicle evades collisions, reduces excessive steering, and avoids sudden braking, where $\lambda_c$, $\lambda_s$, and $\lambda_b$ are assigned values of $0.001$, $0.1$, and $0.1$ respectively.

### B.3 The hyper-parameters of RL training

The hyper-parameters of RL training are elaborated in Tables A7 and A8, and they are same to those in [17] and [20].

Table A7: RL training hyper-parameters on CARLA.

| Hyperparameter | Value |
|---|---|
| Image size | $128 \times 128$ |
| Stacked frames | 3 |
| Action repeat | 1 |
| Batch size | 128 |
| Discount factor $\lambda$ | 0.99 |
| Init steps | 1,000 |
| Episode length | 1,000 |
| Learning algorithm | Soft Actor-Critic (SAC) |
| Number of frames | 100,000 |
| Replay buffer size | 100,000 |
| Optimizer (encoder, actor, critic) | Adam ($\beta_1 = 0.9$, $\beta_2 = 0.999$) |
| Optimizer (transition and reward prediction network) | Adam |
| Learning rate (encoder, actor, critic) | 1e-3 |
| Learning rate (transition and reward prediction network) | 1e-3 |
| Learning rate ($\alpha$ in SAC) | 1e-3 |
| Transition and reward prediction network update frequency | 1 |
| Actor update frequency | 1 |
| Critic target update frequency | 2 |

Table A8: RL training hyper-parameters on DMControl.

| Hyperparameter | Value |
|---|---|
| Image size | $84 \times 84$ |
| Stacked frames | 3 |
| Action repeat | 4 |
| Batch size | 128 |
| Discount factor $\lambda$ | 0.99 |
| Init steps | 1,000 |
| Episode length | 1,000 |
| Learning algorithm | Soft Actor-Critic (SAC) |
| Number of frames | 500,000 |
| Replay buffer size | 100,000 |
| Optimizer (encoder, actor, critic) | Adam ($\beta_1 = 0.9$, $\beta_2 = 0.999$) |
| Optimizer (transition and reward prediction network) | Adam |
| Learning rate (encoder, actor, critic) | 1e-3 |
| Learning rate (transition and reward prediction network) | 1e-3 |
| Learning rate ($\alpha$ in SAC) | 1e-3 |
| Transition and reward prediction network update frequency | 1 |
| Actor update frequency | 1 |
| Critic target update frequency | 2 |

### B.3.1 Hardware Details

**Computing Resources.** All models are trained using a server that is equipped with 4 NVIDIA GeForce RTX 3090 GPUs and a 64-core AMD EPYC 7H12 2.6GHz CPU Processor.

## C Potential Negative Societal Impacts

While DDM offers significant advancements in reinforcement learning by effectively integrating multi-modal data, it is essential to consider and address some potential negative societal impacts. DDM models could be susceptible to adversarial attacks, where malicious inputs are designed to deceive the system into making incorrect decisions. Such attacks could lead to failures in critical applications, such as autonomous vehicles or healthcare systems, potentially causing harm to individuals.

