# OpenReview forum: "Seek Commonality but Preserve Differences: Dissected Dynamics Modeling for Multi-modal Visual RL"
_NeurIPS.cc/2024/Conference — NeurIPS 2024 poster_

### Official Review · Reviewer_xBoX · 2024-07-09

**Soundness:** 3
**Presentation:** 3
**Contribution:** 2
**Rating:** 6
**Confidence:** 4

**Summary:**

The paper proposed a method, namely Dissected Dynamics Modeling (DDM), for multi-modal environment dynamics modeling in visual RL. The core idea is to adopt additional modules to extract separate modality-consistent and modality-inconsistent features in each modality stream with designated losses as the regularization. During training, the model tries to maximize mutual information between modality-consistent features from different modalities at the current and the next timestamp. For modality-inconsistent features from, the model tries to enforce orthogonality. Experiments on CARLA and DMControl indicated the superiority of DDM over other state-of-the-arts and further analyses suggested the effectiveness of DDM v.s. DeepMDP. Ablation studies supported some of the important design choices.

**Strengths:**

- The general idea of attempting to decouple modality-consistent and modality-inconsistent features to improve environment modeling makes sense, which could provide some insights to future work to this particular field

- Experiment results show good improvements over some state-of-the-arts, suggesting DDM's effectiveness for environment modeling in visual RL

- Good ablations, analyses and visualizations provided a comprehensive understanding of the method to some extent justifying DDM's soundness and supporting its superiority for environment modeling in visual RL

**Weaknesses:**

- The proposed method is only showed to be working on a subset of modalities that are visual only, which greatly limits its generalizability and contribution as a method for multi-modal learning. However, visual RL is not my expertise and thus I cannot evaluate the contribution on this particular field.

- Following above, from the experiments it seems that the method also only works on visual inputs from the same camera perspective. i wonder if there are experiments demonstrating the model also works for visual inputs from multiple camera positions.

- For experiments on DMControl, how is the masking done for frames at different timestamps? Also, besides 20% masking, do the authors have results with other masking ratio?

- Some places are not written in a technically accurate way, though it's hard to tell whether it is because of technical misunderstanding or simply misuse of words. For example:
  - Line 46 - 48, "Firstly, modality-correlated features provide a foundational perspective by capturing shared and complemented information across different sensory inputs." The modality-consistent features that are shared among modalities does not COMPLEMENT each other. Instead, those modality-inconsistent features that are unique for each modality themselves complement other modalities.
  - Line 50-51, "these inconsistencies are typically deemed less critical and are filtered out through modality alignment". Modality alignment should not have the effect of filtering out modality-inconsistent features, in my understanding. If the authors have some evidence, I'd like to see.

**Questions:**

Please refer to the above Weaknesses part and address my concern accordingly

**Limitations:**

Yes, limitations are discussed.

---

> ### Author Rebuttal · Authors · 2024-08-07
>
> We are truly grateful for your thoughtful remarks and experimental suggestions. These remarks shed light on what we can improve and are crucial for refining our work. We address your main concerns as follows:
>
> > The method is only tested on visual modalities, limiting its generalizability and contribution.
>
> We appreciate the comment and recognize the importance of generalizing to different modality types. We think our method’s contributions are twofold: the fundamental idea and the method itself. While our method mainly focuses on visual modalities, the core idea—that modality-correlated and distinct features are both crucial—might also provide initial spark of inspiration in non-visual fields. For instance, consider a photo of a sea and a corresponding text description like “Vibrant sea under a clear blue sky, with fluffy white clouds.” The photo may show details that the text misses, such as distant ships. Similarly, the text may also capture high-level concepts that lack in the image feature, such as the sensory concept of “fluffy”. These distinct features between modalities can be valuable for the task at hand, which is worth exploring and might motivate further research on image-text learning.
>
> We hope the potential inspiration brought by our central idea can benefit the non-visual fields and help address some of the concerns. We are thankful for your reflective remark, which suggests a promising extension of our work to fit other modalities (e.g., audio, text) and benefit the entire multi-modal research community. We sincerely appreciate this insight and are committed to work on these extensions.
>
>
> > Experiments for visual inputs from multiple camera positions?
>
> Thanks for the experiment advice. We have further verified the ability of our model on multiple camera positions. First, we switch the camera view of RGB modality in DMControl. Second, we test on CARLA with RGB and LiDAR BEV as input modalities. LiDAR BEV is a bird view map, whose perspective is very different from RGB. Due to the limited time budget, we only compare our method with the most competitive methods on these two environments.
>
> The results and illustrations of different camera views are presented in Table 1,2 and Fig. 1(e),(f) in the rebuttal PDF. These results show that our method also works on multiple camera positions.
>
> > How is the masking done? Do the authors have results with other masking ratio?
>
> Our masking operation is performed independently at each timestamp. Therefore, both the masked modality type and the masking locations vary randomly across different timestamps, which simulates a challenging occlusion scenario.
>
> For other masking ratios, we further test the ratio of 0%, 40%, 60%, and compare our method with SPR, the most competitive baseline on DMControl. The new results are in Table 2 of the rebuttal PDF. The results show that as the masking ratio increases, both SPR and our DDM experience performance drops. However, DDM still outperforms SPR at different ratios.
>
> > Some places are not written accurately, such as Line 46 - 48 and Line 50-51.
>
> We apologize for the confusion caused. For the first issue in Line 46 – 48, we confirm it was a misuse of the word "complemented," which inaccurately described modality consistencies. We meant to convey that the modality-consistent features contain shared and common information and create a unified description for the environment. We appreciate the detailed attention to this error and will correct it.
>
> For the second issue in Line 50-51, our use of the phrase “filter out” may have been overly definitive. We intended to convey that modality alignment aims to encourage consistency and, consequently, mitigate inconsistencies. To clarify this, we provide evidence both in the literature and with experiments. Specifically, we discuss several alignment methods as follows:
>
> **Imposing consistency constraints** is a straightforward method for modality alignment, typically achieved by minimizing cross-modality feature distances [1,2]. Ideally, if two modality features have zero distance, it means perfect consistency and the absence of inconsistencies. We reference multiple works as evidence to support this design goal, as quoted below:
>
> “... the states expressed by different modalities *can be consistent* at the same time... to achieve this, we use a similarity measurement...” (Sec. IV.A in [1])
>
> “...to *ensure the consistency* of the latent embeddings of different modalities in the shared latent space, we develop two parallel cross-alignment schemes...” (Sec.3.2 in [2])
>
> **Mutual information optimization [3]** is also designed to enhance feature consistency (and thereby decrease inconsistencies), we quote [3] as follows:
>
> “The key innovation is a specially-designed density ratio estimator that *encourages consistency* between the latent codes of each modality.” (Abstract in [3])
>
> For experimental evidence, we calculate the CKA index [4] between modality features. CKA is a metric to quantify similarity between network features for the same input. A higher CKA indicates greater feature correlation. Specifically, for RGB modality on CARLA, we extract features $z$ from a baseline SAC model without modality alignment, and the modality-consistent feature $\overline{z}$ and inconsistent feature $\hat{z}$ from our DDM. We Then compare the CKA between these features across 1K input samples. The results are as follows:
>
> ① CKA($z$,$\overline{z}$)=0.865
>
> ② CKA($z$,$\hat{z}$) = 0.897
>
> ③ CKA($\overline{z}$,$\hat{z}$) = 0.561
>
> It can be seen the both ① and ② are relatively high, showing that without modality alignment, $z$ retains both common and unique modality information. Further, ③ is notably lower than ②, indicating that compared with the non-aligned feature $z$, the inconsistent information in the aligned $\overline{z}$ is indeed reduced.
>
> References:
>
> [1,2,3] correspond to [32,27,4] in our paper
>
> [4] Similarity of neural network representations revisited, ICML 2019

---

> ### Comment · Reviewer_xBoX · 2024-08-09
>
> I appreciate the authors' responses, which addressed my concerns well. After reading other reviews and responses, I decided to update my rating from Borderline Accept to Weak Accept.
>
> Please make necessary modification and add the relevant clarification, discussion, and experiment results to your final paper.

---

> > ### Author Response · Authors · 2024-08-10
> > **Thank you for the positive support**
> >
> > We appreciate the comments that provide new perspectives on both experimental design and future research directions. We are glad to hear that your concerns have been addressed satisfactorily. We will ensure that the discussed changes and results are integrated into the revised manuscript to enhance its clarity and robustness. Thank you again for your positive support.

---

### Official Review · Reviewer_dbSG · 2024-07-17

**Soundness:** 3
**Presentation:** 3
**Contribution:** 3
**Rating:** 6
**Confidence:** 3

**Summary:**

The paper presents a solution for better multimodal dynamic modeling in visual RL. The paper claims that existing works only emphasize consistent (aligned) information across modalities, leaving out the opportunity for the model to benefit from the inconsistent features. The work introduces a new consistency loss, where the (cross-modal) mutual prediction happens dynamically, meaning a modality has to infer the other modality's features for the next step rather than the current one. Furthermore, the work introduces a "soft objective" for cross-modal feature orthogonalization. The work shows consistent improvement over the existing state-of-the-art on CARLA and DMControl benchmarks.

**Strengths:**

1) The paper is well-motivated, well-written, and easy to follow. The state-of-the-art results provide stats with several runs. The design of each loss component is straightforward.
2) The paper achieves significant improvement over the reported state-of-the-art results.
3) The cross-modal transition prediction loss is interesting and results in the most significant performance boost, as shown in Table 3. Overall, each loss component is shown to benefit the model's performance.
4) The method is generalizable beyond two modalities.

**Weaknesses:**

1) The authors report the episode to return and driving distance but do not report DS/RC/IP, which is reported in other methods mentioned in the papers and could be more informative for evaluating driving performance. Is there any reason why the authors do not report those metrics?

2) L265-267 states that   $ \mathcal{L_{fp}} $ and $ \mathcal{L}_{r} $ do not benefit much to the existing DeepMDPandSPR. Are there any supporting references/results for this claim?

**Questions:**

1) Do the authors think that incorporating a component of their modeling, e.g., $ \mathcal{L_{tp}} $, into existing models could result in a boost?

2) In Section 3.2 [L166], the authors suggest replacing a stronger objective (Eq. 6) with Eq. 7. Did you observe in your experiments that, indeed, a model trained with Eq.7 is better than Eq.6?

**Limitations:**

The authors discussed potential limitations.

---

> ### Author Rebuttal · Authors · 2024-08-07
>
> Thank you so much for the time and effort invested in reviewing our work. The positive remarks are truly appreciated, and we feel encouraged by the feedback. We address the points raised in the comments as follows:
>
> > The authors report the episode to return and driving distance but do not report DS/RC/IP, is there any reason why the authors do not report those metrics?
>
> Thanks for the thoughtful question on the metrics. Our evaluation protocol on CARLA follows the commonly adopted one in many existing RL works [1,2,3,4]. Different from the setting in TransFuser [5] (which reports DS/RC/IP), the RL agent's goal in this protocol is to drive as far as possible in limited timesteps without colliding into other moving vehicles or barriers. Each episode (i.e., evaluation trial) immediately ends when the agent vehicle collides or reaches the timestep limit. There is no predefined route during evaluation, so RC cannot be calculated. Because each episode immediately ends after a collision, the calculation of IP also becomes impossible. Since RC and IP are both impractical to be obtained, DS cannot be provided. Instead, the RL agent’s performance is primarily evaluated by Episode Reward (ER), which accounts for driving distance and driving stability, such as less abrupt steering.
>
> Note that although we compare with TransFuser in our paper, we evaluate it under the RL protocol rather than its original protocol. As we explained in the paper, this involves integrating the modality fusion and alignment modules of TransFuser with a baseline RL algorithm (SAC). The original setting in TransFuser and the RL setting in our work represent distinct approaches to training autonomous driving agents. The evaluation protocols and metrics for these two settings are also different, which is why we did not report DS/RC/IP scores.
>
> [1] SECANT: Self-Expert Cloning for Zero-Shot Generalization of Visual Policies, ICML 2021
>
> [2] Learning Better with Less: Effective Augmentation for Sample-Efficient Visual Reinforcement Learning, NeurIPS 2023
>
> [3] Model-Based Reinforcement Learning with Isolated Imaginations, TPAMI 2023
>
> [4] Pre-training Contextualized World Models with In-the-wild Videos for Reinforcement Learning, NeurIPS 2023
>
> [5] TransFuser: Imitation with Transformer-Based Sensor Fusion for Autonomous Driving, TPAMI 2023
>
> > L265-267 states that $\mathcal{L}\_{fp}$ and $\mathcal{L}\_r$ do not benefit much to the existing DeepMDP and SPR. Are there any supporting references/results for this claim?
>
> Thanks for identifying this issue. There might be a misunderstanding here. Specifically, we are not trying to use $\mathcal{L}\_{fp}$ and $\mathcal{L}\_r$ to benefit DeepMDP and SPR. In L265-267, we state “However, this enhancement does not bring a significant advantage over conventional methods such as DeepMDP and SPR”. We mean that although $\mathcal{L}\_{fp}$ and $\mathcal{L}\_r$ can improve the performance of the baseline model, the improved results are still not significantly better than the results of DeepMDP and SPR (reported in Table 1 of our paper). So, we are describing a cross-table comparison, by comparing the $+\mathcal{L}\_{fp}$ and $+\mathcal{L}\_r$ rows in Table 3 of our paper and the DeepMDP and SPR rows in Table 1 of our paper. We did not mention Table 1 in L264-267, which may cause confusion. We appreciate your detailed examination and will correct this issue in the revised paper.
>
> > Do the authors think that incorporating a component of their modeling, e.g., $\mathcal{L}\_{tp}$, into existing models could result in a boost?
>
> We are confident that our modeling such as cross-modality transition prediction $\mathcal{L}\_{tp}$ can boost existing methods. This is because our method is flexible and is not limited to any particular RL model or network architecture. To verify the effectiveness of our method on other models, we further apply $\mathcal{L}\_{tp}$ to DrQ [1] and evaluate its performance. The results are given in Table 2 of the rebuttal PDF, which show that $\mathcal{L}\_{tp}$ can also boost the performance of existing RL models.
>
> [1] Image augmentation is all you need: Regularizing deep reinforcement learning from pixels, ICLR 2020
>
> > In Section 3.2 [L166], the authors suggest replacing a stronger objective (Eq. 6) with Eq. 7. Did you observe in your experiments that, indeed, a model trained with Eq.7 is better than Eq.6?
>
> Yes, we have compared Eq.6 and Eq.7 in Sec.4.4 (L304-L313) of our paper. The results are presented in Fig.7 of our paper, which show that models trained with Eq.7 consistently perform better than models trained with Eq.6.

---

> > ### Comment · Reviewer_dbSG · 2024-08-12
> >
> > Dear authors,
> > I greatly appreciate your responses and the additional results presented in the PDF.
> > I think the authors addressed all my comments and I think this work should be accepted.

---

> > > ### Author Response · Authors · 2024-08-13
> > > **Appreciation for the supportive feedback and recommendation**
> > >
> > > We are truly grateful for your positive comments and for recognizing the efforts in our response and additional results. Thank you once again for your support and the time you invested in reviewing our work thoroughly.

---

### Official Review · Reviewer_Msqh · 2024-07-22

**Soundness:** 3
**Presentation:** 3
**Contribution:** 3
**Rating:** 6
**Confidence:** 4

**Summary:**

The paper proposes Dissected Dynamics Modeling (DDM), a dynamics modeling framework for learning latent features in multi-modal visual RL. The methodology focuses on capturing both the shared and distinct information contained across input modalities. The paper presents a multi-modal architecture and training loss designed to accomplish this, and experimentally demonstrates the benefits of DDM on tasks from CARLA and the DeepMind Control Suite.

**Strengths:**

**[S1] Novel approach to multi-modal RL:** The paper proposes a novel methodology for multi-modal visual RL that accounts for both the similarities and differences across input modalities, compared to prior works that largely focus only on the similarities across input modalities. Experiments demonstrate the benefits of this novel approach.

**[S2] Detailed experimental analysis:** The paper provides detailed experimental analysis of the proposed DDM method. This includes comparisons across several types of baselines (although some choices of baselines are not state-of-the-art), ablation studies that justify design choices, robustness analysis, and visualizations for qualitative understanding.

**[S3] Important problem:** The paper focuses on how to effectively leverage information from multiple input modalities for decision making. This addresses an important consideration for the successful deployment of RL in real-world systems where multi-modal data is common.

**Weaknesses:**

**[W1] Clarity of implementation / experimental details:** The high-level idea of the DDM framework is clear, but some of the implementation details are not clearly described. Some components of the experiments were also not described in detail, which made it difficult to interpret some of the results. Please see the questions below.

**[W2] Connections to related work:** DDM appears to build upon previous works in visual / multi-modal RL for its dynamics modeling and consistency extraction components, but these connections are not made clear. I think it would be useful to better emphasize what parts of DDM build upon existing works vs. what parts are novel contributions. (i) The dynamics modeling approach looks similar to reconstruction-free approaches in visual RL [a,b,c,d], but the similarities / differences are not discussed. (ii) It is mentioned that existing multi-modal approaches have focused on aligning modalities, but the paper does not discuss how the proposed consistency extraction method relates to these approaches.

**[W3] Heuristic design choices:** Design choices of the proposed architecture are justified through experimental results, but no theoretical support is provided (or connections to existing approaches that provide theoretical support).

---

References:

[a] Gelada et al. (2019). DeepMDP: Learning continuous latent space models for representation learning.

[b] Schwarzer et al. (2021). Data-efficient reinforcement learning with self-predictive representations.

[c] Okada et al. (2021). Dreaming: Model-based reinforcement learning by latent imagination without reconstruction.

[d] Zhang et al. (2021). Learning invariant representations for reinforcement learning without reconstruction.

**Questions:**

**Methodology:**

**[Q1]** Are gradients with respect to the latent feature encodings $z$ stopped in any components of the loss function in (13), or are they propagated through the loss function everywhere that $z$ appears?

**[Q2]** Are the next-step target values $z_{t+1}$ in (5) and (11) calculated using the same feature encoder used for $z_t$? It would be helpful to make this more clear. Are gradients taken through $z_{t+1}$ in (5) and (11)?

---

**Experiments:**

**[Q3]** What do the single modality baseline results represent? Do they only use RGB images, or do they consider a combined input that incorporates information from all modalities? If these baselines are not restricted to RGB images, it would be useful to include results (baselines or ablation of DDM) using only RGB inputs to demonstrate that the use of multi-modal inputs leads to improved performance over standard visual RL.

**[Q4]** DreamerV3 [e] and DrQ-v2 [f] are strong model-based and model-free visual RL algorithms, respectively. Why were DeepMDP and SPR chosen as baselines instead of DreamerV3 (which has reported better head-to-head performance vs. SPR in Atari 100k)? Why was DrQ chosen as a baseline instead of DrQ-v2 (which has reported significant improvements over DrQ on DeepMind Control Suite)?

**[Q5]** Has performance converged by the end of training for the experimental results, given the number of training steps used (100k and 500k in CARLA and DeepMind Control Suite, respectively)? In prior works on visual RL, DeepMind Control Suite tasks are typically trained for longer than this (1M steps in DreamerV3 [e], 3M steps in DrQ-v2 [f]), and these baselines had not converged to final performance after 500k steps.

---

**Minor:**

Spelling: Inon. --> Incon. (p. 2, Figure 1 caption)

---

References:

[e] Hafner et al. (2023). Mastering diverse domains through world models.

[f] Yarats et al. (2021). Mastering visual continuous control: Improved data-augmented reinforcement learning.

**Limitations:**

The authors have addressed the limitations of the current work.

---

> ### Author Rebuttal · Authors · 2024-08-07
>
> We would like to express our sincere gratitude for the insightful comments. We also deeply appreciate the suggestions regarding presentation and experimentation. Below are our responses to your concerns:
>
> > [W1] Some implementation and experiment details are not clear.
>
> Please refer to our responses to [Q1-Q5] for detailed clarification.
>
>
> > [W2] Connections to related work need to be emphasized.
>
> Thank you for the suggestion. Here we discuss the connections in detail:
>
> **(i) Connections with dynamics modeling methods [a-d]**
>
> The primary similarity between [a-d] and our DDM is the state and reward prediction strategy commonly used in RL dynamics modeling. However, DDM differs as it is tailored for multi-modal RL, introducing novel approaches in both modality decomposition and dynamics modeling.
>
> Specifically, both DeepMDP [a] and DDM predicts latent states and rewards. However, DDM does not address modality relationships or learn state transitions in a modality-aware manner. SPR [b] uses multi-step state prediction but it overlooks modality interactions. Dreaming [c] mainly focuses on learning over different timesteps and samples. Differently, DDM aims at dissecting features across different modalities. DBC [d] employs a bisimulation metric to model dynamics, which differs from our dissected modeling strategy.
>
>
> **(ii) Relation to existing consistency extraction methods**
>
> Although the goal of drawing modality features closer for consistency extraction is similar between DDM and existing methods, the techniques vary significantly. MAIE [32] directly minimizes modality features at the same timestep. In contrast, DDM predicts cross-modality transition across adjacent timesteps. TransFuser [5] and EFNet [40] utilize attention, MUMMI [4] optimizes mutual information, and HAVE [20] employs attention and hypernetworks. Like MAIE, these methods do not create a synergy between dynamics modeling and modality alignment, which is a unique feature of the cross-modality transition prediction in our DDM.
>
> (Reference numbers are the same as in our paper)
>
> > [W3] No theoretical support is provided.
>
> Thank you for emphasizing the need for a theoretical analysis. We have conducted initial research to find theoretical insights that support our approach.
>
> Our investigation suggests that information theory [1] might provide a suitable theoretical framework for our study. For instance, research in [2] shows how multiple sources provide structured multivariate information about a target variable. Analogously, in multi-modal RL, each modality can be treated as a source and the agent's action as the target. Based on this, we aim to establish a lower bound on the information gap between using only modality-consistent features versus all available information. Deriving this lower bound will underscore the importance of both modality consistencies and inconsistencies in decision-making, thus supporting our method design and empirical findings.
>
> However, despite our earnest efforts on this idea, establishing a definitive result remains challenging for us within the limited rebuttal timeline. We sincerely value your feedback and will deepen this theoretical analysis in future work.
>
> [1] The Mathematical Theory of Communication, Bell System Technical Journal, 1948
>
> [2] Nonnegative Decomposition of Multivariate Information, arXiv, 2010
>
>
> > [Q1] Are gradients of $z$ stopped in (13)?
>
> Sorry for not clarifying this. We stop the gradient of all prediction target features in (4), (5), (8), and (11). That is, for each loss term of the form $||X(z_a)-z_b||^2$, where $X$ represents prediction head and $z_a$, $z_b$ are modality features, we stop the gradient of $z_b$.
>
> > [Q2] 1. Are $z_{t+1}$ in (5) and (11) used the same encoder as $z_t$? 2. Are gradients taken through $z_{t+1}$?
>
> 1. Yes, they are. We do not utilize a moving average encoder like in SPR [b] but rather use the same online encoder while stopping the gradient of the target values like in DBC [d].
>
> 2. The gradients are stopped as explained in Q1. This avoids potential model collapse, which is observed in SPR paper when no stop gradient operation is applied to the target values.
>
> ([b,d] are the same as in [W2])
>
> > [Q3] What are single modality baseline results? Any results using only RGB inputs?
>
> The results consider a combined input of all modalities. Following the advice, we have tested the case where only RGB inputs are used. The results are in Table 1 in the rebuttal PDF, which shows that the baseline with multi-modal input indeed outperforms RGB.
>
> > [Q4] Why use DeepMDP, SPR and DrQ as baselines instead of DreamerV3 and DrQ-v2?
>
> For single-modality baselines, we would like a direct comparison of the dynamics modeling technique itself and reduce the impact of other factors (e.g., the RL algorithm and the use of planning mechanism or not). Therefore, we choose DeepMDP, SPR and DrQ, all use SAC as RL algorithm without planning, just like our method. We appreciate the recommendation of DreamerV3 and DrQ-v2 and have conducted experiments with them. As shown in Table 2 of the rebuttal PDF, DrQ-v2 obtains stable performance, while DreamerV3 gets inferior results on the Cartpole swingup_sparse task. This is probably caused by the sparse reward and modality inconsistencies, which hampers the learning of its transition model.
>
> > [Q5] Has the performance converged?
>
> Our setting of training steps follows DBC [d] and SECANT [1]. To verify the convergence, we further train different methods with double steps (i.e., 200K/1M for CARLA/DMControl). The results are given in Fig.1 (a)-(d) in the rebuttal PDF. The figure shows that the improvement beyond 100K/500k steps becomes exceedingly marginal. Therefore, our current training can achieve a satisfactory level of performance.
>
> [d] is the same as in [W2]
>
> [1] SECANT: Self-Expert Cloning for Zero-Shot Generalization of Visual Policies, ICML 2021
>
> > Minor: Inon.-> Incon.
>
> Thanks for spotting this, we will correct it.

---

> > ### Comment · Reviewer_Msqh · 2024-08-08
> > **Response**
> >
> > Thank you for the detailed responses and additional experimental results. Incorporating some of these discussions / results into the paper will improve clarity and further strengthen the experimental analysis. My main questions and concerns have been addressed, and I have increased my overall review score to reflect this.

---

> > > ### Author Response · Authors · 2024-08-09
> > > **Thank you for the supportive feedback**
> > >
> > > We appreciate the guidance provided in your detailed and informative review. We will incorporate these discussions and results into our revised manuscript. Thank you again for your supportive feedback.

---

### Author Rebuttal · Authors · 2024-08-07

We sincerely appreciate all reviewers for the valuable time and effort dedicated to reviewing our work. The comments have been highly constructive, and the positive evaluations from all reviewers are immensely encouraging. We have carefully considered each remark and have responded accordingly. For a quick and convenient overview, we briefly outline our responses as follows:

**1. Additional experiments**. Following the reviewers’ suggestions, we have conducted additional experiments, such as evaluating more baselines, employing our design to other RL models, and test on different camera perspectives.

**2. Clarification of details**. We have clarified several details regarding our technical design and paper presentation, such as the gradient flow during training and the explanations of certain words and phrases in our paper.

**3. Further discussions**. Based on the reviewers’ comments, we have conducted additional discussions on several key aspects, such as the relationship of our method with existing methods and the analysis of modality alignment methods.

Please see our detailed responses below and the attached rebuttal PDF for addressing the individual comments from each reviewer. In addition to the rebuttal, we will incorporate these responses into our revised paper to further enhance its clarity. We thank you once again for your insightful feedback.

---

### Decision · Program_Chairs · 2024-09-25

**Decision:**

Accept (poster)

**Comment:**

This paper proposes a multimodal Visual RL setting where the input modalities contain similar informative cues in both modalities while may also have inconsistent cues, and the goal is how to capture these disparate features for effective policy learning. The paper approaches this problem from a disentanglement perspective, and proposes to project the consistent cues to a common subspace, while the inconsistent cues between the modalities are disentangled using a mutual orthogonality constraint.

The paper received positive reviews overall, with three weak accept recommendations. While, there have been some concerns initially on i) the lack of theoretical foundations for the approach, ii) lack of clarity in implementation details, iii) experimental analysis, and iv) relations to prior works, authors provided a strong rebuttal addressing the concerns. AC concurs with the reviewers judgement that paper is well-crafted and demonstrates promising results and recommends acceptance. Authors should include the additional technical details that came up in the rebuttal into the final paper.

Further, the current literature review in the paper is quite limited and is specific to visual RL. There is a large body of works on disentangled representation learning, which is the central idea in this paper and that must be reviewed in detail, especially in the context of cross-modal alignment. Some pointers are given below. Authors should contrast their work to such literature and clarify how the proposed model stands out.
1. Learning to Decompose and Disentangle Representations for Video Prediction, Hsieh et al. NeurIPS 2018
2. Vector-Decomposed Disentanglement for Domain-Invariant Object Detection, Wu et al., CVPR 2021
3. Semantics Disentangling for Generalized Zero-Shot Learning, Chen et al., ICCV 2021
4. SA-DVAE: Improving Zero-Shot Skeleton-Based Action Recognition by Disentangled Variational Autoencoders, Li et al, ECCV 2024